# Does Living with Grandparents Affect Children’s and Adolescents’ Health? Evidence from China

**DOI:** 10.3390/ijerph182211948

**Published:** 2021-11-13

**Authors:** Jian Yu, Zhonggen Zhang

**Affiliations:** China Academy for Rural Development, Zhejiang University, Hangzhou 310058, China; 11822021@zju.edu.cn

**Keywords:** grandparents’ coresidence, children and adolescent health, childhood obesity, nutritional balance

## Abstract

The level of nutritional health of children and adolescents is an important indicator of social development, and grandparents, as common co-caregivers, may have a considerable impact on the health level of their grandchildren. In this paper, we investigated the effects of grandparents’ coresidence on children’s and adolescents’ nutritional health levels using the CHNS (China Health and Nutrition Survey) database using the PSM (Propensity Score Matching) method’ and identified heterogeneity in the effects of grandparents’ coresidence by PSM grouping in terms of urban and rural areas, age, and the mothers’ education level. It was found that grandparents’ coresidence is beneficial for children’s and adolescents’ health to a certain extent. Overfeeding and spoil that many people worried when grandparents involved in childcare did not happen in our samples. Moreover, grandparents’ coresidence had a more significant effect on the health level of children and adolescents in rural areas at a younger age and with a lower level of maternal education. Government and families should put more effort into equipping caregivers with knowledge on how to raise their grandchildren better.

## 1. Introduction

The level of children’s and adolescents’ nutritional health, as an essential indicator of the country’s comprehensive development level and health status, has been one of the major concerns of research in the field of food nutrition and health and the field of rural human capital in recent years. Children and adolescents’ health problems are still prevalent worldwide today and cannot be ignored. The “Healthy China 2030” Initiative states that it is necessary to highlight the health problems of key populations, such as children [1], and strengthen interventions to achieve health protection from the beginning of pregnancy to the end of life and maintain health of the people comprehensively. Children’s health status is closely related to their health, income, education level, and productivity in adulthood, and poor nutritional health negatively affects their future education level, productivity, and labor income, which is detrimental to the formation of human capital [2,3,4]. Although the incidence of global hunger has declined dramatically since 1992, various nutritional health problems, including traditional undernutrition, micronutrient deficiencies, as well as overweight and obesity, are still prevalent among children and adolescents in all countries [5,6].

Much previous work has focused on children’s and adolescents’ health problems, but the effect of grandparents’ coresidence is under-analyzed. In addition to genetic factors, children’s and adolescents’ health is influenced by numerous factors such as eating habits, dining environment, and lifestyle, which are determined mainly by their caregivers, especially in the early stages of growth [7,8]. In recent years, with the increase in women’s labor force participation, grandparents living with their grandchildren and acting as the primary or co-parent of them, i.e., intergenerational care, has become an option for many families, and this phenomenon is widespread at home and abroad, in rural and urban areas [9]. Statistics show that the primary caregiver for children aged 0–5 years in their daily lives and the daily educator of the family is firstly mothers, followed by the grandparents [10]. In rural areas, it is more common for left-behind children to be cared for by their grandparents, and the physical development and nutrition levels of non-left-behind children may be higher than the level of left-behind children [11].

The reasons why grandparents’ coresidence may affect children’s health levels are as follows. First, many grandparents lack proper health knowledge, especially in rural areas, and grandparents tend to believe that the more children eat and the fatter they are, the healthier they are, thus overfeeding children and causing them to become obese. Second, the traditional diet in many areas of China is unhealthy, and grandparents are more likely to uphold traditional feeding habits than parents, leading to nutritional imbalances in children and adolescents and thus leading to vitamin deficiencies, anemia, and other health problems. Moreover, third, many grandparents in China today grew up in a time of material scarcity, and they are more likely to spoil their children than the parents and satisfy children’s demands for all kinds of snacks, which are often high in sugar and fat, thus quickly leading to childhood obesity. Fourth, grandparents may encourage children and adolescents to participate in physical activity less because they are older and less likely to accompany them and more likely to protect them from injury, which may also lead to overweight children [12,13,14].

This paper focuses on whether grandparents’ coresidence has a significant effect on children’s and adolescents’ nutritional health level and whether the effect of grandparents’ coresidence is heterogeneous in terms of urban–rural, age, and mother’s education level. Coresidence here means that the child in a family has at least one grandparent living together and at least one parent in the meantime. This paper focuses on children’s and adolescents’ health in five dimensions: whether they are overweight, whether they are malnourished, dietary preferences, disease status, and exercise status. First, we use the Propensity Score Matching Method (PSM) to measure the extent of the effect of grandparents’ coresidence on their grandchildren’s health, and second, we use the PSM grouping method to examine the heterogeneity of this effect. This paper examines the impact of grandparents’ coresidence on children’s and adolescents’ health and well-being from the perspective of grandparents’ coresidence, which can not only deepen the knowledge of the advantages and disadvantages of intergenerational care, a common phenomenon at home and abroad, extend the research perspectives in the field of rural human capital and expand the content of empirical studies but also provide references for micro-families to adjust their child care practices, as well as provide a basis and inspiration for the formulation of public service policies related to rural human capital.

## 2. Literature Review and Innovation Points

From a combination of domestic and international studies, the main factors affecting children’s and adolescents’ health can divide into three categories: biological factors, health-related behavioral factors, and socioeconomic factors. Biological factors include natural biological conditions of children and adolescents such as parental genes, innate physical fitness, gender, age, and maternal age at birth [9,15], for example, whether mothers are overweight or not may influence children’s habits and health level through channels such as genetics and similar lifestyles [12]. Health-related behavioral factors include behaviors related to children’s daily lives, such as nutritional intake and physical activities. It is generally believed that children and adolescents who spend more time in daily physical activity have higher health levels [16]. Socioeconomic factors can be further divided into two significant aspects: family and community. On the family side, the educational level of parents or primary caregivers, the economic status of the family, and the number of children may affect children’s nutritional health. The existing studies on the effect of mothers’ educational level on children’s health are controversial: on the one hand, mothers with higher education levels are more likely to engage in non-agricultural work and spend less time on childcare, which has a negative impact on children’s health [9]; on the other hand, more educated mothers may be more knowledgeable about childcare and thus have a beneficial effect on children’s health [12,17]. Many studies show that higher parental income is associated with better child nutritional health. Only children, especially girls, have a more pronounced advantage in nutritional health [9]. At the community level, the social environment in which children and adolescents live, such as the level of regional economic development and the quality of the regional environment, also affects the level of their health, for example, access to primary health care services, including maternity services, essential drugs, and immunization has dramatically improved children’s health status [18]; the improvement of infrastructure, such as water access in rural communities, also has an important impact on children’s health status [18,19,20,21]; children and adolescents living in urban and suburban areas have significantly better health status than those living in counties and villages; children and adolescents in both eastern and central regions have significantly better health status than those in the western areas [9].

Children’s and adolescents’ nutritional health has specific measures, such as whether they have good physical development, meet various nutritional indicators, or suffer from multiple diseases. The indicators that have been used more often in research generally include height-for-age scores (the standard recommended by the World Health Organization) [22] and children’s body mass index (BMI) [23].

Among the many factors that influence children’s and adolescents’ nutritional health, grandparents’ coresidence is a factor that has received less attention in existing studies but is of critical importance. From the perspective of children development, among the above three categories of factors affecting children’s and adolescents’ nutritional health, the first category of biological factors is inherent and unchangeable, the second category of health-related behavioral factors is primarily influenced by the third category of family and community factors, and community factors are also difficult to change at the micro-level. Therefore, it is most feasible to improve the related family factors to improve children’s growth and health, and the family factors are mainly reflected by the caregivers. Consequently, this paper focuses on intergenerational coresidence as the primary concern.

In the existing literature on the effects of grandparents’ coresidence on children’s and adolescents’ health, domestic studies mainly focus on the issue of children left behind in rural areas, and the findings are divergent, with some studies concluding that children’s nutritional health is better when accompanied by their parents [4,9,24,25], and some hold the view that children’s health will not differ when they are cared by parents or grandparents [26]. This same disagreement exists in evidence from other countries, with one view suggesting that grandparents have a disadvantage compared to parents in caring for children [12,13,14,27], and another that children raised by relatives, such as grandparents, do not differ significantly in health levels from those raised by parents [28]. The positive and negative effects of grandparents’ care mentioned in the above studies are listed below:(1)Positive effects: As grandparents are involved in the care of children, mothers have more employment opportunities, household income will increase, and thus the nutritional status of children will improve. Moreover, some studies suggest that children raised solely by grandparents are not significantly different from children raised in traditional families with two biological parents.(2)Negative effects: Children cared solely by grandparents have a higher risk of illness and malnutrition and face poorer medical conditions, including access to health care and health insurance.

Evidence from other countries on factors influencing the decision of grandparents’ coresidence has concluded that this decision is mainly influenced by two significant factors: economic and cultural factors, as well as other aspects such as whether the parents are widowed or not, and their education level. Specifically in China, in terms of culture, due to the influence of traditional cultural concepts, grandparents generally regard intergenerational parenting as the responsibility of family succession and take caring for their grandchildren as a necessary “task” in their old age, and grandparents’ decision to live together may vary from region to region due to the considerable differences in customs and traditions. In terms of economy, parents can hardly take care of and educate their children on their own due to employment and schooling, as they have to survive and develop in a rapidly industrializing and modernizing society and provide better material resources for their children, while grandparents are idle at home due to retirement, which makes intergenerational care by grandparents living together gradually become the main form of family education. In the meantime, due to the presence of a large number of migrant workers in rural areas, many rural children and adolescents are separated from their parents for long periods, and it is widespread for children and adolescents to live with their grandparents, so families in rural areas with poorer economic conditions are more likely to have grandparents to care for their children. In addition, families with lower educational attainment and widowed mothers are more likely to choose grandparents’ coresidence [29,30,31,32,33].

The primary measurement methods used to study the effects of grandparents’ coresidence on children’s and adolescents’ nutritional health include econometric methods such as multiple regression, propensity score matching (PSM), difference-in-difference method (DID), and instrumental variable method (IV), which can effectively address potential endogeneity while estimating the effects of intergenerational care very well. For example, Sun (2016) used the PSM-DID method to evaluate the impact of parental work away from home on the health of children left behind [24].

Among the existing studies on the effects of grandparents’ coresidence on children’s and adolescents’ health levels, relevant evidence from other countries focuses on the impact of grandparents’ coresidence on children’s obesity without addressing other health indicators, and domestic literature focuses on rural left-behind children under grandparental care, with measures such as current health levels and children’s body size. In general, there is still room for improvement in the existing studies: first, the existing studies have a single measure of children’s and adolescents’ health, lacking careful consideration of malnutrition, overweight and obesity, nutritional balance, and exercise health. Second, the existing studies lack classification discussions, generally only looking at the children and adolescents in the data as a whole but rarely discussing children and adolescents in different groups in-depth, while their grandparents may influence different groups. Third, the existing studies mainly focus on the situation of grandparents taking care of children instead of their parents but pay less attention to the phenomenon of joint care by both, which is more common nowadays. Fourth, there are few recent studies, and most of the data seems old, so the validity of the information is weak; fifth, the existing findings on the impact of grandparental care on children’s and adolescents’ health are still debated. Accordingly, this paper uses data from the China Health and Nutrition Survey (CHNS) database from 1991–2015 to measure children’s and adolescents’ health by selecting five dimensions (overweight and obesity, malnutrition, physical fitness, nutritional balance, and length of exercise) to discuss the effect of grandparents’ coresidence on children’s and adolescents’ health and to provide a comparison by classifying children and adolescents by variables such as urban and rural areas, children’s age, and mothers’ education level. Compared with previous studies, this paper provides a more comprehensive measure of children’s and adolescents’ health in the area of grandparents’ coresidence, discusses the effects of grandparents’ coresidence on children and adolescents in different situations in greater depth, has better timeliness, and is a valuable addition to existing studies.

## 3. Methods, Data, and Variables

### 3.1. Empirical Method

Combining previous studies, children’ s and adolescents’ health model constructed in this study is: (1)Y=βX+e
where Y denotes the health level of children and adolescents, X denotes the explanatory variables, including individual factors, family characteristics, community environment, and grandparents’ coresidence, and e is the error term and follows a normal distribution.

To measure the effect of grandparents’ coresidence on children’s and adolescents’ nutritional health levels, the following econometric model is developed:(2)$$Y_{it}=\alpha +\delta D_{it}+\beta X_{it}+\varepsilon_{it}$$

In Equation (2), i denotes different individual children and adolescents, $Y_{it}$ represents the level of children’s and adolescents’ nutritional health, $D_{it}$ denotes whether individual i’s household has grandparents living with him/her at year t, $X_{it}$ is other explanatory variables, α is a constant term, and $\varepsilon_{it}$ is a random disturbance term. If samples are randomly assigned to the coresidence group or non-coresidence group, δ captures the extent to which grandparents’ coresidence affects children’s nutritional health. However, there may be an individual self-selection problem in whether households choose to live with children’s grandparents, i.e., it may be the result of households’ conscious choice according to their specific economic conditions, family status, etc. Suppose the above model is directly adopted without considering the potential self-selection problem. In that case, the explanatory variables are correlated with the random perturbation term, there is an endogeneity problem, and the error term and parameter estimates of the results may be biased.

The propensity score matching (PSM) method is a common method to deal with the self-selection problem so as to measure the effect level more accurately. In this paper, we apply the propensity score matching method to deal with the self-selection problem of grandparents’ coresidence in each sample, i.e., based on the non-cohabitant samples, we match one (or more) non-cohabitant samples for each cohabitant sample, so that the matched samples have basically the same characteristics except whether they choose grandparents’ coresidence or not. The difference of the outcome variable between the two matched samples is the average treatment effect on the treated (ATT), which is expressed as:(3)$$ATT=E\left(Y_{1t}|D=1\right)-E\left(Y_{0t}|D=1\right)=E\left(Y_{1t}-Y_{0t}|D=1\right)$$

In Equation (3), $Y_{1t}$ is the nutritional health level of children and adolescents when they live with their grandparents and $Y_{0t}$ is the nutritional health level of children and adolescents when they do not live with their grandparents, where $E\left(Y_{0t}|D=1\right)$ is not actually observable, and this counterfactual result will be estimated by the PSM method.

After that, in order to detect differences in the effect of grandparents’ coresidence on children’s and adolescents’ health levels in different situations, PSM grouping was used to divide the total sample into two subsamples with subgroup variables such as whether they were located in rural areas before grouping them for PSM again, then the PSM results were compared for each subsample.

### 3.2. Data Source

The data used in this paper are obtained from the China Health and Nutrition Survey (CHNS) database. The database was established by the University of North Carolina at Chapel Hill and the Chinese Center for Disease Control and Prevention (CCDC), and covers about 7200 households and more than 30,000 people in 15 provinces and cities in China for ten years, including 1989, 1991, 1993, 1997, 2000, 2004, 2006, 2009, 2011, and 2015. The CHNS database uses a multistage, random cluster process to draw the sample surveyed in each province. Counties in the nine provinces were stratified by income (low, middle, and high), and a weighted sampling scheme was used to randomly select four counties in each province. Villages and townships within the counties and urban/suburban neighborhoods within the cities were also selected randomly. Additionally, the CHNS database provides detailed information on household relationships, economic level, health data, and dietary data, making the data comprehensive and representative. The subsamples selected in this paper are the information related to children and adolescents aged 2–18 years with complete information on parents and grandparents living together in the nine survey years from 1991 to 2015. Since few consecutive samples in the database meet the screening conditions, the subsamples in the article only have cross-sectional data. Children and adolescents here mean any individual aged 2–18 years old in our sample.

### 3.3. Variable Selection

According to the definition of children’s and adolescents’ nutritional health in the previous paper [34,35,36,37], this paper has five outcome variables:(1)Whether children and adolescents are overweight(2)Whether children and adolescents are malnutritioned(3)Whether children and adolescents have been sick in the last four weeks(4)Nutritional balance(5)Length of exercise

The first and the second indicators are selected to measure the overall development level of children and adolescents. The other three outcome variables are designed according to the influence mechanism of grandparents’ coresidence on children’s health to test whether the influence mechanism holds. Among them, indicators (1) and (2) are judged according to existing mainstream standards. Children aged 2–6 years are grouped according to the weight for age Z-score (WAZ) index score calculated in the 2013 China Public Health Statistical Yearbook; children with a Z-score greater than two are considered overweight, and those with a Z-score less than −2 are considered malnourished; the criteria for overweight and malnutrition in children and adolescents aged 7–18 years are calculated as children’s BMI and then classified according to the Classification Standards for Overweight and Obesity Screening Body Mass Index Values in Chinese School-Age Children and Adolescents [38] and the Screening Standards for Malnutrition in School-Age Children and Adolescents [39], respectively. Children’s BMI is the body mass index of children and adolescents, and the internationally used formula for calculating BMI is
(4)$$BMI=Weight/Height^{2}$$

Indicator (3) refers to whether children and adolescents have been ill or injured or have suffered from chronic or acute diseases in the past four weeks. Indicator (4) is whether children’s and adolescents’ daily diet has low protein or high fat intake, calculated according to children’s and adolescents’ three-day average intake, and the measurement standard refers to the Chinese Dietary Nutrient Reference Intakes 2013 Revised Edition [40]. Indicator (5) is the sum of children’s and adolescents’ weekly exercise duration in and out of school (unit: minutes).

In the selection of matching variables for PSM and PSM grouping, this paper first selects variables based on theories and existing studies, then selects and adjusts matching variables based on the matching effect, and finally selects matching variables for PSM, which are listed below:(1)Gender(2)Children’s age(3)Do children live in the rural area(4)The number of children in the household(5)Whether the household drinking water source is running water(6)Whether the mother is divorced(7)The mother’s BMI(8)The region and the time factors
among which (1) and (2) are biological factors, household factors include variables from (3) to (7), and the region and the time factors represent socioeconomic factors.

The PSM grouping variables are as below:(1)Children’s age(2)Do children live in the rural area(3)Mothers’ education level

Considering the sample size and matching effect, all samples are divided into two sample groups in this paper for each grouping variable. The “children” and “adolescents” are separated by 12 years old (as 12 is the legal age for children to graduate from primary school and go to middle school in China); “high level” in the variables of mothers’ educational level indicates high school education or above, and “low level” indicates junior high school education or below.

## 4. Empirical Results

### 4.1. Descriptive Analysis of the Data

Table 1 shows the essential information and detailed explanation of all the variables involved in this paper. As can be seen from the mean values in Table 1, in terms of children’s and adolescents’ health, about 13.2% of the sample are overweight, about 12.6% are malnourished, about 6.5% are sick and injured, have chronic or acute illnesses in the last four weeks, about 50.1% are nutritionally unbalanced, and the average number of minutes of exercise per week is about 288 min. In terms of sample characteristics, the male and female samples are more evenly split, each household have an average of 1–2 children, 65.6% of households use tap water or bottled water as drinking water, about 1.8% of the sample have divorced or widowed (widowed) parents, and the average BMI of mothers is 22.5. Among the grouping variables, 72.5% of the samples are from rural areas, the average age of the samples is around ten years old, and the mothers’ education levels are concentrated in junior high school and below.

Table 2 provides preliminary descriptive statistics for the outcome and explanatory variables of interest in this paper divided into two groups. It can be roughly seen that children and adolescents living with grandparents may be more likely to be overweight, more likely to have a recent injury or chronic illness, and may spend more time exercising, while no effect has been observed on malnutrition and nutritional balance in children with or without living with grandparents. Children living with grandparents are younger, tend to live in the city, are more likely to be the only child and have tap water as drinking water in the home, and their mothers have a lower BMI and are more likely to be divorced or widowed. There is little difference in gender between the two sample groups.

### 4.2. Estimation of Grandparents’ Coresidence Decision Equation

Table 3 shows the results of the logit model estimation of the grandparents’ coresidence decision equation. The outcome variables in columns (1)–(5) are Overweight, Malnutrition, Ever sick or injured (4 weeks), Nutrition balance, and Physical exercise (per week) in that order. Considering the combined effect of both logit models with coresidence as the dependent variable and the PSM results with five items (Overweight et al.) as the outcome variable, the matching variables entering the five models are slightly different. The regression results in Table 3 are generally consistent with the perceptions and hypothesized expectations of existing studies. Younger children, city families, families with fewer children, families with access to tap water, and mothers with a lower BMI are more likely to live with grandparents.

### 4.3. Measurement of the Effect of Grandparents’ Coresidence on Child Health

Level changes, ATT values, and t-statistics for the five nutritional health outcome variables are reported in Table 4, Table 5, Table 6, Table 7 and Table 8, respectively. Four matching methods are used for changes in each outcome variable to minimize sample loss, and the estimates for all four matching methods are relatively close. The mean results obtained using the four matching methods show that grandparents’ coresidence affects children’s and adolescents’ overweight and obesity, malnutrition, sickness, nutritional balance, and exercise to varying degrees. Specifically, compared to children and adolescents who do not live with their grandparents, the ones who lived with their grandparents have a 2.5% decrease in overweight, a 1% increase in the probability of recent illness or chronic disease, a 3.9% increase in nutritional balance, and an average of 20 more min of exercise per week. This result reflects that grandparents’ coresidence affects children’s and adolescents’ physical health slightly and that negative and positive effects coexist, which requires further specific discussion.

Up to this step, it can be assumed that the overall effect of grandparental residence on children’s health is minimal. Table 4, Table 5, Table 7 and Table 8 show a slight positive effect of grandparents’ coresidence on children’s and adolescents’ health, with minimal changes in the odds of being overweight, obese, or malnourished when children and adolescents live with their grandparents and a small but significant increase in nutritional balance and hours of physical exercise. Although Table 6 shows a higher probability of recent illness among children with grandparents living with them, the value is only 1%.

These results confirm the claims of previous studies that children cared for by both grandparents and parents are virtually indistinguishable from children cared for by only their parents in terms of health. Children and adolescents living with grandparents even perform better in body health, nutrition balance, and physical exercise hours. This may be because parents, especially mothers, get more time and more opportunities to engage in work, and household income increases, and thus the nutritional status of children and adolescents improves.

### 4.4. Balancing Test Results

To test the validity of the matching results, after matching the samples, this paper further tests the statistical significance of the differences in the matching variables between the two groups of samples, and the balancing test results are given by Table 9, Table 10, Table 11, Table 12 and Table 13. Because the sample sizes and some of the variables entering the five nutritional health models are different, the balance results of all five outcome variables are reported in this paper. After sample matching, the standardized deviations of the explanatory variables are within 2%; the *p*-values of the likelihood ratio tests indicate that the joint significance tests of the matched variables are statistically significant before matching and largely rejected after matching; the Pseudo R^2^ values also decreased significantly to 0.000~0.002 after matching. The results of the above tests show that the sample matching successfully balances the distribution of the explanatory variables between the two groups of samples.

### 4.5. Heterogeneity Analysis of the Effect

The results of the heterogeneity analysis measured by the PSM grouping method based on nearest neighbor matching (1–5 matching) are shown in Table 6.

Row (1) in Table 14 shows that there are urban–rural differences in the effect of grandparents’ coresidence on children’s and adolescents’ health, with a more significant positive impact on children and adolescents in rural areas. It can be seen that the phenomenon of grandparents’ coresidence in urban areas only slightly affects whether children and adolescents are overweight and have a balanced diet, while the other outcome variables do not pass the significance test. In contrast, the outcome variables of children and adolescents in rural areas are all significantly affected by grandparents’ coresidence except the Malnutrition variable. This result may be related to the inherent differences between rural and urban areas, where human resource endowment is at an overall disadvantage, and grandparents’ involvement compensates for the shortfall in endowment by increasing the number of childcare providers, which makes the health condition of children and adolescents better. In addition, some health indicators may be more strongly subject to effective interventions at the school and community levels in urban areas, and thus, the effect of grandparents’ coresidence is not significant.

Row (2) in Table 14 shows that grandparental residence has a health effect only for younger children, while none of the five outcome variables are significant for the adolescents’ group. This difference may be because after the age of 12, most children enter junior high school and begin to have a greater sense of autonomy in their eating and other habits, and some children start to live in school (“zhu xiao”), thus the family influence on their health is gradually reduced.

Row (3) shows the difference of mothers’ education level on grandparents’ coresidence effect. The results show that grandparents’ coresidence will help more in nutrition balance when mothers have a lower education level or in physical exercise when mothers have a higher education level. That is maybe because children’s nutrition conditions become better when more people get involved in childcare, and highly educated mothers may be busier, so grandparents will offer more time to accompany with children.

Overall, the effect of grandparents’ coresidence on children’s and adolescents’ health is influenced by urban-rural, children’s age, and paternal human capital endowment. Children and adolescents with more superior social and human capital endowments have less influence from grandparents’ coresidence; younger children are more likely to be influenced.

## 5. Conclusions

Using CHNS data, this paper first applies the PSM method to measure grandparents’ coresidence on children’s and adolescents’ health using five outcome variables, which are estimated to find that grandparents’ coresidence may contribute to children’s and adolescents’ overfeeding, nutrition, and exercise. The matching effects are then tested for balance and found to be plausible. Finally, the heterogeneity of the effect of grandparents’ coresidence on children’s and adolescents’ health is estimated separately for the three dimensions of urban and rural areas, children’s age, and mothers’ education using the PSM grouping method. It is found that the effect of grandparents’ coresidence on children and adolescents is more remarkable in rural areas and more significant on younger children, and it offers more help in nutrition balance when mothers are lower educated or in physical exercise when mothers are higher educated.

According to the use of large sample data and reliable PSM methods, our study provides new evidence on the impact of grandparents’ coresidence on children’s and adolescents’ health. It has implications for both micro-family decision making and macro-policy development. Our findings show that grandparents’ coresidence is beneficial for children’s and adolescents’ health to a certain extent. Overfeeding and spoiling that many people worry about when grandparents are involved in childcare did not happen in our samples. In a large number of Chinese families with two working parents, grandparents’ coresidence can be an effective aid to childcare. In addition, there are some shortcomings in our study. Firstly, CHNS does not include school-level data, which is considered an essential variable influencing children’s and adolescents’ health, so we cannot control for differences in school conditions between urban and rural areas in this study. Secondly, we have not considered the situation that grandparents live in the same city or county with children and adolescents, which is a larger group of samples and may have different influences resulting from coresidence.

Despite the limitations of this paper, two recommendations can be made based on the findings.

For individuals and families, firstly, when parents do not have enough time and energy to take care of their children, it is an excellent choice to ask for some help from grandparents if possible, and grandparents’ involvement contributes to children’s and adolescents’ health. Secondly, 13% of children and adolescents in our study were overweight, and 50% did not have a balanced nutrition condition. Parents need to pay more attention to children’s nutrition balance and body weight at a young age since it is essential for children to develop good eating habits, and families influence it mainly in early childhood. Thirdly, parents need to offer more companionship and care to their children when they are the only caregivers.

For policymakers, firstly, children’s and adolescents’ health problems in rural areas and families with low human capital endowment cannot be ignored. Children and adolescents in the care of fewer caregivers may be disadvantaged in several dimensions. There is a need to strengthen education on diet and health knowledge in rural and backward areas through social institutions such as schools, neighborhood committees, and village committees to compensate for the lack of family care. Secondly, the nutritional balance of Chinese children is generally at a low level. The government needs to intervene in this area for the sake of the future quality of the population. There are many ways that the government can achieve these two goals. For example, regular activities such as dietary and nutritional knowledge popularization can be provided in hospitals, and free physical fitness and nutritional development testing for children and adolescents can be provided at the community level.

## Figures and Tables

**Table 1 ijerph-18-11948-t001:** Variables’ description and descriptive statistics.

Variables	Variables Description	Mean	Std. Dev.
**Outcome Variables**			
Overweight	Yes = 1, No = 0	0.132	0.339
Malnutrition	Yes = 1, No = 0	0.126	0.332
Ever sick or injured (4 weeks)	Ever sick or injured last 4 weeks; Yes = 1, No = 0	0.065	0.246
Nutrition balance	Ever nutrition imbalance; Yes = 1, No = 0	0.501	0.500
Physical exercise (per week)	Total minutes of physical exercise per week	287.999	267.732
**Matching variables**			
Gender	Male = 0, Female = 1	0.471	0.499
Age	Children’s and adolescents’ age	10.142	4.628
Living in rural area	Living in rural area = 1, Others = 0	0.725	0.446
Num children	Num children surveyed and not surveyed in the household	1.652	1.297
Water	How household obtain drinking water; Tap/bottled water = 1, Others = 0	0.656	0.475
Mother divorced or widowed	Whether mother was divorced or widowed during the investigation	0.018	0.132
Mother’s BMI	Mother’ s BMI index	22.497	2.940
**Grouping variables**			
Mother’s education level	None = 0, Primary school = 1, Lower middle school degree = 2, Upper middle school degree = 3, Technical or vocational degree = 4, University or college degree = 5, Master’s degree or higher = 6	1.637	1.299

Note: The variables “Age” and “Living in rural area” are not listed repeatedly as they are both matching variables and grouping variables.

**Table 2 ijerph-18-11948-t002:** Descriptive statistics and two-sample *t*-tests for the difference in means.

Variables	(1)	(2)	(3)	(4)	(5)
	Without Grandparents’Coresidence		With Grandparents’ Coresidence		Difference
	N	Mean	N	Mean	
**Outcome variables**					
Overweight	7899	0.130	5248	0.163	−0.033 ***
Malnutrition	7856	0.124	5224	0.127	−0.003
Ever sick or injured (4 weeks)	8867	0.061	5760	0.081	−0.021 ***
Nutrition balance	7835	0.500	4826	0.512	−0.012
Physical exercise (per week)	2102	279.394	1725	305.115	−25.721 ***
**Matching variables**					
Gender	8988	0.469	5864	0.467	−0.002
Age	8988	10.404	5864	8.580	1.824 ***
Living in rural area	8988	0.743	5864	0.695	0.048 ***
Num children	8988	1.752	5864	1.289	0.463 ***
Water	8988	0.627	5864	0.709	−0.082 ***
Mother divorced or widowed	8988	0.014	5864	0.016	−0.002
Mother’s BMI	8988	22.530	5864	22.320	0.210 ***

Note: *** represent a 1% significance level. Grandparents’ coresidence is defined as living with at least one grandparent. It is a binary variable.

**Table 3 ijerph-18-11948-t003:** Logit models with coresidence.

Dependent Variable: Coresidence	(1)	(2)	(3)	(4)	(5)
Gender	0.019 (0.037)		0.026 (0.035)	0.011 (0.038)	0.105 (0.069)
Age	−0.084 *** (0.004)				
Living in rural area	−0.176 *** (0.044)	−0.200 *** (0.041)	−0.075 * (0.041)	−0.167 *** (0.044)	−0.056 (0.076)
Num children	−0.085 *** (0.024)		−0.199 *** (0.022)	−0.179 *** (0.023)	−0.341 *** (0.077)
Water	0.092 ** (0.044)		0.074 *** (0.041)	0.077 * (0.044)	0.015 (0.103)
Mother divorced or widowed	0.109 (0.153)	−0.033 (0.151)	−0.034 (0.143)	−0.118 (0.155)	0.563 ** (0.258)
Mother’s BMI	−0.025 *** (0.006)	−0.041 *** (0.006)	−0.042 ** (0.006)	−0.035 ** (0.007)	−0.028 ** (0.011)
Region control	YES	YES	YES	YES	YES
Time control	YES	YES	YES	YES	YES
Constant	0.595 *** (0.163)	0.107 *** (0.151)	0.445 *** (0.153)	0.325 *** (0.166)	0.070 (0.429)
Pseudo R2	0.0572	0.0317	0.0365	0.0280	0.0729
LR chi2	1011.24 ***	557.63 ***	715.62 ***	470.84 ***	383.90 ***
Number of observations	13147	13080	14627	12661	3827

Note: *, **, and *** represent a 10%, 5%, and 1% significance level, respectively. Region controls in models from (1) to (4) are designed as three regional dummy variables in China: eastern, central, and western provinces, which are separated by average economic development levels, and model (5) uses regional fixed effect directly.

**Table 4 ijerph-18-11948-t004:** Effect of grandparents’ coresidence on children and adolescents whether overweight.

Matching Methods	Overweight			
	Coresidence	Non-Coresidence	ATT	T-Statistics
Nearest neighbors matching (1–5)	0.163	0.193	−0.030 ***	−4.06
Nearest neighbors matching (1–10)	0.163	0.192	−0.030 ***	−4.14
Kernel matching (bandwidth = 0.06)	0.163	0.185	−0.022 ***	−3.22
Kernel matching (bandwidth = 0.10)	0.163	0.179	−0.016 **	−2.42
Mean	0.163	0.187	−0.025	−3.46

Note: ** and *** represent a 5%, and 1% significance level, respectively.

**Table 5 ijerph-18-11948-t005:** Effect of grandparents’ coresidence on children’s and adolescents’ malnutrition.

Matching Methods	Malnutrition			
	Coresidence	Non-Coresidence	ATT	T-Statistics
Nearest neighbors matching (1–5)	0.127	0.112	0.015 **	2.14
Nearest neighbors matching (1–10)	0.127	0.116	0.011 *	1.67
Kernel matching (bandwidth = 0.06)	0.127	0.120	0.007	1.10
Kernel matching (bandwidth = 0.10)	0.127	0.121	0.006	0.95
Mean	0.127	0.117	0.010	1.47

Note: * and ** represent a 10% and 5% significance level, respectively.

**Table 6 ijerph-18-11948-t006:** Effect of grandparents’ coresidence on children’s and adolescents’ recent illness.

Matching Methods	Ever Sick or Injured (4 Weeks)			
	Coresidence	Non-Coresidence	ATT	T-Statistics
Nearest neighbors matching (1–5)	0.081	0.071	0.010 **	2.02
Nearest neighbors matching (1–10)	0.081	0.071	0.010 **	2.15
Kernel matching (bandwidth = 0.06)	0.081	0.072	0.009 **	1.97
Kernel matching (bandwidth = 0.10)	0.081	0.071	0.010 **	2.23
Mean	0.081	0.071	0.010	2.09

Note: ** represent a 5% significance level.

**Table 7 ijerph-18-11948-t007:** Effect of grandparents’ coresidence on children’s and adolescents’ nutrition balance.

Matching Methods	Nutrition Balance			
	Coresidence	Non-Coresidence	ATT	T-Statistics
Nearest neighbors matching (1–5)	0.512	0.554	−0.042 ***	−4.04
Nearest neighbors matching (1–10)	0.512	0.556	−0.043 ***	−4.35
Kernel matching (bandwidth = 0.06)	0.512	0.550	−0.038 ***	−4.06
Kernel matching (bandwidth = 0.10)	0.512	0.543	−0.031 ***	−3.33
Mean	0.512	0.551	−0.039	−3.95

Note: *** represent a 1% significance level.

**Table 8 ijerph-18-11948-t008:** Effect of grandparents’ coresidence on children’s and adolescents’ activity.

Matching Methods	Physical Exercise (Per Week)			
	Coresidence	Non-Coresidence	ATT	T-Statistics
Nearest neighbors matching (1–5)	304.302	284.293	20.009 *	1.94
Nearest neighbors matching (1–10)	304.302	286.496	17.806 *	1.79
Kernel matching (bandwidth = 0.06)	304.302	281.959	22.343	2.34
Kernel matching (bandwidth = 0.10)	304.302	282.693	21.608	2.29
Mean	304.302	283.860	20.441	2.09

Note: * represent a 10% significance level.

**Table 9 ijerph-18-11948-t009:** Balancing test results for independent variables before and after matching (outcome variable: Overweight).

Matching Methods	Pseudo R2	LR chi2	*p*-Value	MeanBias (%)	MedBias (%)
Unmatched	0.057	1000.57	0.000	12.6	10.6
Nearest neighbors matching (1–5)	0.002	20.02	0.074	1.8	1.9
Nearest neighbors matching (1–10)	0.002	20.45	0.108	1.8	1.9
Kernel matching (bandwidth = 0.06)	0.001	18.11	0.382	1.5	1.1
Kernel matching (bandwidth = 0.10)	0.001	16.84	0.465	1.4	0.9

**Table 10 ijerph-18-11948-t010:** Balancing test results for independent variables before and after matching (outcome variable: Malnutrition).

Matching Methods	Pseudo R2	LR chi2	*p*-Value	MeanBias (%)	MedBias (%)
Unmatched	0.032	556.67	0.000	9.3	9.9
Nearest neighbors matching (1–5)	0.001	18.75	0.131	1.7	1.4
Nearest neighbors matching (1–10)	0.001	16.36	0.230	1.8	1.5
Kernel matching (bandwidth = 0.06)	0.001	7.41	0.880	1.1	0.7
Kernel matching (bandwidth = 0.10)	0.001	19.04	0.122	1.6	1.1

**Table 11 ijerph-18-11948-t011:** Balancing test results for independent variables before and after matching (outcome variable: Recent illness).

Matching Methods	Pseudo R2	LR chi2	*p*-Value	MeanBias (%)	MedBias (%)
Unmatched	0.036	713.37	0.000	11.1	10.1
Nearest neighbors matching (1–5)	0.000	7.74	0.956	1.0	1.0
Nearest neighbors matching (1–10)	0.001	9.90	0.872	1.1	1.0
Kernel matching (bandwidth = 0.06)	0.000	7.15	0.970	1.0	0.8
Kernel matching (bandwidth = 0.10)	0.001	19.96	0.222	1.6	1.0

**Table 12 ijerph-18-11948-t012:** Balancing test results for independent variables before and after matching (outcome variable: Nutrition balance).

Matching Methods	Pseudo R2	LR chi2	*p*-Value	MeanBias (%)	MedBias (%)
Unmatched	0.028	468.72	0.000	9.9	7.0
Nearest neighbors matching (1–5)	0.001	14.65	0.477	1.6	1.2
Nearest neighbors matching (1–10)	0.001	10.58	0.782	1.2	1.4
Kernel matching (bandwidth = 0.06)	0.000	3.88	0.998	0.8	0.6
Kernel matching (bandwidth = 0.10)	0.001	15.99	0.382	2.0	1.7

**Table 13 ijerph-18-11948-t013:** Balancing test results for independent variables before and after matching (outcome variable: Physical exercise).

Matching Methods	Pseudo R2	LR chi2	*p*-Value	MeanBias (%)	MedBias (%)
Unmatched	0.073	382.21	0.000	11.1	8.2
Nearest neighbors matching (1–5)	0.001	5.89	1.000	1.5	1.7
Nearest neighbors matching (1–10)	0.001	5.13	1.000	1.3	1.4
Kernel matching (bandwidth = 0.06)	0.001	2.44	1.000	0.9	0.7
Kernel matching (bandwidth = 0.10)	0.001	3.96	1.000	1.2	1.1

**Table 14 ijerph-18-11948-t014:** Results of heterogeneity analysis.

Outcome Variables		Overweight		Malnutrition		Ever Sick or Injured (4 Weeks)		Nutrition Balance		Physical Exercise	
		ATT	T-Stat	ATT	T-Stat	ATT	T-Stat	ATT	T-Stat	ATT	T-Stat
**(1)**	**Urban**	−0.027 *	−1.87	−0.001	−0.07	0.004	0.37	−0.039 **	−2.12	8.319	0.44
	**Rural**	−0.022 **	−2.48	0.014	1.62	0.013 **	2.39	−0.040 ***	−3.18	31.322 **	2.50
**(2)**	**Adolescents**	−0.009	−0.85	0.007	0.53	−0.001	−0.08	−0.019	−0.97	4.343	0.24
	**Children**	−0.032 ***	−3.47	0.014 *	1.68	0.006	0.96	−0.047 ***	−3.80	18.133	1.47
**(3)**	**High level**	−0.036 *	−1.89	0.009	0.72	0.009	0.76	−0.043 *	−1.95	50.560 **	2.45
	**Low level**	−0.016 **	−1.99	0.009	1.13	0.004	0.68	−0.049 ***	−4.19	17.198	1.44

Note: *, **, and *** represent a 10%, 5%, and 1% significance level, respectively.

## Data Availability

Publicly available datasets were analyzed in this study. This data can be found here: https://www.cpc.unc.edu/projects/china/data/datasets (accessed on 27 September 2021).

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
