# Peer review of "Does Living with Grandparents Affect Children’s and Adolescents’ Health? Evidence from China"

_ijerph, 2021, doi:10.3390/ijerph182211948_

Round 1

Reviewer 1 Report

The manuscript focuses an interesting subject in the field of public health, i.e., the influence of living with grandparents on children's health status in five dimensions (overweight/obesity, malnutrition, physical fitness, nutritional balance, and physical activity level). The authors analyze data from Chinese population in the period from 1991 to 2015 using propensity score matching, showing evidence of heterogeneous effects of grandparents in children's health.

The subject of the manuscript is original, seeking to explore the information available on health and nutrition surveys with Chinese children throughout significant period of time to verify the role of grandparents in human capital formation. The evidence produced in the study will be useful to support the design of public policies that promote children's health status through changes in lifestyle, also improving health outcomes during adult life in the long run.

Nevertheless, there are several points for improvement of the manuscript, further described below.

  1. General remarks: the manuscript requires extensive revision of English language. The text is very truncated in certain parts and include several strange expressions (e.g., “pay more efforts” instead of “put more efforts” in line 18, “fetus” instead of “beginning of pregnancy” in line 28, “workability” instead of “productivity” in line 31, “human social capital” instead of “human capital” in line 32, “foreign studies” and “foreign literature” instead of “evidence from other countries” in lines 118 and 159, “not reasonable” instead of “unhealthy” in line 141, among others).
  2. Abstract: it is important to include the meaning of the initials CHNS and PSM, which are only mentioned in the Methods.
  3. Introduction: revise the structure and the arguments of the Introduction, which presents only fragments of information focusing the context and the justification of the study.
  4. Literature Review: on the other hand, the literature review encompasses compelling evidence from diverse studies that could be synthesized to support the arguments in the Introduction of the manuscript.
  5. Methods:

5.1. The description of the section is mostly appropriate; however, Equation 1 does not seem to contribute to the general explanation of the models because it represents a generic equation on children's health model, which is further substituted in the study by Equation 2.

5.2. In addition, the inclusion of notes on the Results seems inappropriate within the Methods, in lines 201-202: “(this can also be confirmed by the results in Table 2 later)”; therefore, I suggest to exclude the sentence.

5.3. The description of the data source lacks indication on the survey design. i.e., whether the China Health and Nutrition Survey (CHNS) refers to cross sectional or longitudinal information of individuals.

5.4. The definition of children adopted in the study design is very strange, encompassing individuals from 2 to 18 years-old, which includes also adolescents, according to the World Health Organization classification (people between 10 and 19 years-old). Although the Convention on the Rights of the Child considers that “child means every human being below the age of eighteen years”, I would advise the authors on either including “children and adolescents” in the title and in the definition of the study, or excluding the adolescents from the analysis performed in the model. Considering the differences in behavior and independency between children and adolescents, it would be important to emphasize that the study involved both groups.

5.5. In addition, the explanation on the reasons for exclusion of children under 2 years-old is very confusing, it would be important to revise the text and the arguments.

5.6. The authors state that “According to the definition of children's nutritional health in the previous paper” (line 240); however, there is no reference to the paper cited.

5.7. The description of variables selected for analysis is presented in one single paragraph, which makes difficult to identify each variable separately, thus, I suggest to present each variable in a numbered list.

5.8. The equation to estimate the Body Mass Index should be included in a single paragraph separated from the main text, otherwise, it seems very odd.

5.9. Tables 1 and 2 should be included in Results, encompassing descriptive analysis of the data in the sample.

5.10. Authors did not include year of the survey among matching variables, it would be important to indicate the reason.

5.11. Additionally, indicate whether there are differences in the characteristics of the population according to year of the survey.

  1. Results:

6.1. The description of the results in Table 3 are very confusing: authors state that “outcome variables in columns (1)-(5) are Overweight, Malnutrition, Ever sick or injured (4 weeks), Nutrition balance, and Physical exercise (per week)”; however, the heading of the table indicates that the dependent variable is “Coresidence” with grandparents, and the title of the table indicates “Logit model with coresidence”.

6.2. Authors do not indicate whether the models estimated include control variables for years of the survey, which should be important to control for effects of economic fluctuations and other potential changes occurring throughout the period of analysis.

6.3. The section includes the description of the results obtained in the models, and comments on the potential mechanisms leading to the results; however, there is lack of proper link with the evidence on the literature. Therefore, I suggest that the authors include comparisons of the results with data from the literature on the subject.

6.4. The first paragraph of the subsection “Heterogeneity analysis of the effect” refers to the description of cutoff points of variables that should be included in the Methods (lines 362-368).

  1. Conclusions:

7.1. The first paragraph of the section comprises a repetition on the main results obtained in the models; however, the text is very truncated and does not explore the interesting findings of the study appropriately.

7.2. The description of the limitations of the study encompasses a single sentence that seems extracted from other study, since it does not connect with the previous paragraph. In addition, the limitations of the study should include the survey design, the potential for bias in the responses from participants, and the potential absence of important observable characteristics that may influence the relationships between grandparents and children, among others.

7.3. The recommendations based on the findings of the study are very interesting; however, the text of the paragraphs should be revised to clarify the points presented by the authors.

Reviewer 2 Report

Revision is needed for the following points.

First, the review of previous studies is not specific. Previous studies should be specifically reviewed. It is necessary to summarize the negative and positive effects on coresidence at a glance in previous studies.
Second, a detailed explanation of the sampling method is required. Representativeness of the sample and error should also be presented. Basic statistical analysis about sociodemographic variables should be presented.
Third, in the analysis contents, it is necessary to examine whether Table 3 is an analysis necessary in the context of the entire study to analyze the determinants of coresidence.
Fourth, the research results are contradictory to each other. A more detailed explanation is needed on how to interpret the positive and negative effects, and the relevance to existing research should be revealed.
Fifth, the conclusion is very short. It is necessary to describe the theoretical and practical contribution of this study in detail. It is necessary to set the discussion and implication sections as independent chapters.

Round 2

Reviewer 1 Report

The manuscript was greatly improved through incorporation of suggestions of the reviewers. The authors incorporated clarifications on issues presented in the first review, and the text is very appropriate. The methods and the resuls were detailed, being adequately presented in the study. There are still minor problems in text editing (e.g., lack of spacing between text and numbers presented), and English language that may be addressed by the authors in the final rounds of revision and publication. In addition, there is a structure of a table that was not properly deleted in page 13 (line 454) of the manuscript. The study is very interesting and may be published after these few corrections.

Author Response

Thank you very much for your careful comments and kind recognition. 

We have added missing spaces between text and numbers, and deleted the table structure in line 454.

This revision also improves several other formatting details and English expressions in the manuscript. Please see the revised version for more information.

Reviewer 2 Report

All of things were well revised

Author Response

Thank you very much for your careful comments and kind recognition. 
This revision has improved several formatting details and English expressions in the manuscript. Please see the revised version for more information.